# Exploration of NIR Squaraine Contrast Agents Containing Various Heterocycles: Synthesis, Optical Properties and Applications

**DOI:** 10.3390/ph16091299

**Published:** 2023-09-14

**Authors:** Shahir Sarasiya, Sara Sarasiya, Maged Henary

**Affiliations:** 1Department of Chemistry, Georgia State University, Atlanta, GA 30303, USA; ssarasiya1@student.gsu.edu (S.S.); ssarasiya2@student.gsu.edu (S.S.); 2Center of Diagnostics and Therapeutics, Georgia State University, Atlanta, GA 30303, USA

**Keywords:** squaraine dye, synthesis, optical properties, near-infrared region, indole, quinoline, perimidine

## Abstract

Squaraine dye is a popular class of contrast near-infrared (NIR) dyes. Squaraine dyes have shown the ability to be modified with various heterocycles. The indole moiety is the most notable heterocycle incorporated in squaraine dyes. A tremendous amount of work has gone into developing indole-based squaraine dyes and determining their applications. The optical properties of squaraine dyes containing an indole moiety facilitate high quantum yields and molar absorptivity, but the absorbance maxima is capped near 700 nm. This is the major limitation of indole-based squaraine dyes. In comparison, other heterocycles with larger conjugated systems such as quinoline and perimidine have demonstrated promising optical properties and immense potential for modifications, albeit with limited development. Quinoline- and perimidine-based squaraine dyes have molar extinction coefficients over 100,000 M^−1^ cm^−1^ and absorbances over 800 nm. This report will look at indole-, quinoline-, and perimidine-based squaraine dyes. Due to the sheer number of reported dyes, the search for indole-based squaraine dyes has been limited to reports from the past five years (2018–2023). For quinoline- and perimidine-based squaraine dyes, a holistic search was performed to analyze the optical properties and applications, due to the abovementioned limitation. This report will evaluate the three different classes of squaraines: indole-, quinoline-, and perimidine-based, to evaluate their optical properties and applications, with the goal of encouraging the exploration of other heterocycles for use in squaraine dyes.

## 1. Introduction

Small molecular organic dyes are a research hotspot. There have been various classes of dyes reported over the years. Squaraine dyes are one of the popular classes of near-infrared dyes that have been described. The unique characteristic that differentiates them from other dyes is the central linking unit; this unit is known a squaraine, where the name for these dyes is obtained. The core unit is composed of an unsaturated, four-membered ring [1,2,3]. This unique linking core unit provides the properties associated with squaraine dyes. Some of their other characteristics are sharp absorption bands, a high molar extinction coefficient, and photoconductivity abilities [4,5,6].

The squaraine core is composed of a four-membered ring that is electron-deficient when incorporated into the dye [7]. The electrons in the squaraine core are delocalized, allowing them to encompass the whole conjugated system. This squaraine core is capped with donor units, forming a donor-acceptor-donor system, allowing the dye to be stabilized by a π-conjugated system [1,7]. The squaraine core scaffold is zwitterionic and has multiple resonances, due to the delocalized electrons [8,9,10]. The zwitterionic nature is an essential part of the dye, as it facilitates electron movement through the scaffold [2,3]. The targeting specificity of the squaraine dyes can be improved through conjugation by synthesizing them with various possible functional groups. The overall optical characteristics of the compound may be affected by the addition of heterocycles and functional groups. Regardless of the varying functional groups and heterocycles, squaraine dyes benefit from excellent chemical and photophysical properties [2,3].

The use of near-infrared squaraine dyes aims to achieve absorbance and fluorescence in the near-infrared region (NIR), and this region ranges from 650 to 1700 nm [11,12]. This region is optimal for biomedical [11,12] and solar energy applications [13]. In terms of biomedical applications, the region enables a higher signal-to-noise ratio for bioimaging. This is due to the limited autofluorescence of biological molecules, resulting in reduced light scattering in this region [14]. Red shifting into the NIR range allows greater penetration of light into tissues, allowing for better spatial visualization [14,15]. Solar radiation is composed of about 50% near-infrared light [16]. To create an efficient solar cell, the light-absorbing materials need to absorb light in this region [17]. In addition, the lower-energy photons of NIR result in a higher short-circuit current density [18].

In recent years, the indole heterocycle has been a popular donor unit for squaraine dyes. There have been numerous applications in various fields reported for this class [4,19,20,21]. In comparison, when looking at other heterocycles, such as quinoline- and perimidine-based squaraine dyes, there are only a few scattered reported dyes and associated applications. This is the case even though they have exciting and notable optical properties that differ from indole-based squaraine dyes.

Herein, the review investigates the three donor units, heterocycles, for squaraine dyes. The three heterocycles are indole, quinoline, and perimidine. When incorporated into the dye, these heterocycles have fascinating reported applications and optical properties. The review looks at the applications and optical properties of notable indole-based squaraines synthesized in the past few years, while the analysis of squaraines containing quinoline and perimidine is explored at a more holistic level, with the ultimate goal of raising awareness of the other unique heterocycles containing squaraine dyes and their reported optical properties and applications.

## 2. Synthesis of Squaraine Dyes

There are two different classes of squaraine dyes: symmetrical and unsymmetrical. The symmetrical squaraine dyes contain identical donor units on each side, while the unsymmetrical, as the name suggests, contain two different donor groups. Typically, the donor units are bound to the first and third positions of the squaraine unit, due to their desirable, red-shifted optical properties compared to the 1,2 regioisomer [22]. For symmetrical squaraine dyes, the synthesis involves the condensation reaction in a single-pot reaction mixture [23], as utilized in Equation (1).

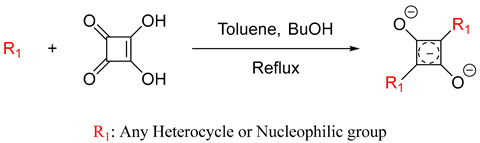
(1)Equation (1): general synthesis of symmetrical squaraine dyes.

Symmetrical squaraine dyes use a standard synthetical process regardless of the heterocycle. The first accepted synthesis of squaraine dyes was reported by Treibs and Jacob in 1965 [24]. The synthesis utilized acetic anhydride as the solvent to form the dye, which is traditionally used for cyanine synthesis. In 1966, a new solvent system was introduced that used the azeotropic principle to produce the squaraine dye; the updated solvent system used an n-butanol–benzene mixture [23]. However, due to the carcinogenic properties of benzene, toluene was utilized instead for the synthesis [25]. This solvent system can azeotropically remove water residue from the reaction and allow a high-temperature system for the condensation reaction. The use of butanol and toluene or benzene (2:1 to 1:1) is the common solvent system for squaraine dye synthesis. In addition, bases such as TEA or quinolone can be incorporated to facilitate the reaction. Most synthetic yields for symmetrical squaraine dyes hover around 60% [26].

Symmetrical squaraine dyes are synthesized with the activation of squaric acid by the butanol or another alkyl alcohol group. This activates the central moiety for the nucleophilic attack by the electron-rich donor unit. This forms the semisquaraine intermediate, which is short-lived, as another nucleophilic attack from the other donor unit follows it [27]. This results in the formation of the dye and the release of water as a byproduct (Figure 1) [27].

The first unsymmetrical squaraine dye was reported in 1968 by Treibs and Jacob, only a couple years after their initial reporting of symmetrical squaraine dye [28]. The synthesis of unsymmetrical squaraine dyes utilizes a different process than its symmetrical counterpart, but mechanistically, they are comparable. The synthesis for unsymmetrical squaraine dye is a multistep process that complicates the synthesis process [29]. For the synthesis, a modified squaric acid is utilized to reduce the reactivity and slow the reaction down. This allows the formation of the semisquaraine product in a higher yield. If the unmodified squaric acid is used, the semisquaraine would be produced in lower yields of the squaraine dye. Modifying the squaric acid includes capping the hydroxyl group with a short alkyl chain or replacing with hydroxyl group with halogens, usually chlorine [30].

The basic synthesis of unsymmetrical squaraine dye entails the formation of the semisquaraine followed by the condensation of the second donor unit with the semisquaraine to form the unsymmetrical squaraine dye (Figure 2). However, before the second condensation occurs, the modified squaric acid needs to contain the hydroxyl group, allowing the semisquaraine to become more reactive for the condensation reaction. This process can be carried out using acid or base hydrolysis [30,31]. The overall yield depends on the donor units. The yield usually ranges from 49% to 20%, but it could be even lower [30]. The lower yields are due to the multiple purification processes needed at each step before proceeding to the next.

Squaraine dyes can have modified squaraine cores incorporated within them. Squaric acid can be modified with different groups such as methoxy, amines, dicyanomethylene, and others [32]. Additions of alkoxy or amino groups can be achieved after the dye has been synthesized [33]. This will result in the formation of the semisquaraine, which can be reacted with the other heterocycle [33].

In addition to forming the dyes using the traditional reaction method, the Dean–Stark apparatus, an alternative method of microwave irradiation, has been used to synthesize the dye. As is known, microwave heating significantly reduces the total reaction time [34]. Barbero et al. reported this to be true for synthesizing symmetrical and unsymmetrical squaraine dyes. In addition, the yield of symmetrical squaraine dyes is higher when the microwave method is used, and similar results are seen for unsymmetrical dyes [26]. The reaction time for unsymmetrical dyes drastically changes, due to acid or base hydrolysis not being needed for the reaction, as shown in Table 1. As seen in Table 1, there is a significant reduction in reaction time for all the dyes, with timing saving of over 300% on the lower end.

## 3. Indole-Based Squaraine Dyes

The indole heterocycle is one of the most common heterocycles used for squaraine dyes. This heterocycle is an aromatic system that consists of a benzene ring fused to a pyrrole ring. This system is an electron-rich heterocycle that contains delocalized electrons [36]. The synthesis of indole is achieved through Fisher indole synthesis. Through this synthesis pathway, a wide variety of indole derivates can be synthesized. In addition, the nitrogen can be alkylated to exponentially increase the total number of indoles synthesized, as presented in Figure 3. When the alkylation of the nitrogen atom occurs, the nitrogen atom obtains a positive charge, forming a salt, iodolium. The positive charge on the nitrogen atom allows the acidic methyl group to react with the squaric acid under basic conditions. The iodolium reacts with squaric acid to form an indole-based squaraine dye (Figure 3). The dye follows the proposed mechanism presented in Figure 1.

### 3.1. Optical Properties of Indole-Based Dyes

The optical properties of selected indole-based squaraine dyes from the past few years show their diverse range of features. The properties depend on the structure of the molecule, i.e., if they are symmetrical or unsymmetrical squaraine dyes. Below are selected molecular structures of symmetrical dyes (Figure 1) and their listed optical properties (Table 2) from the past five years (2018–2022). The optical properties that will be considered are the absorbance maxima (λ_abs_), emission maxima (λ_em_), molar extinction coefficient (ε), quantum yield (φ_f_), and Stokes shift.

When considering the structure of these symmetrical dyes, common locations of modifications are observed. The first region where modification is commonplace is the alkylation of the nitrogen atom on the heterocyclic ring. Based on Figure 1, there is a wide range of modifications from no alkylation—squaraines **5**, **10**, **12**—or the addition the of a peptide, RGD, as seen in squaraine **16**. *N*-alkylation can introduce new charges to the compounds, improving their water solubility, as in squaraines **1**–**4**, **9**, **11** [4,37]. Another common site for modification is the fifth position of the indole. There is an introduction of halogens, sulfonic acid, carboxylic acid, the benzene group or other substituents to the dye at this position. The modification at this position is another location in which to add charge(s) to the compound, which can be achieved by adding sulfonic acid and carboxylic acid groups. The central squaraine core can be modified, as seen in squaraines **9**–**13**. For these dyes, the modification seen is with dicyanomethylene, barbituric acid or other groups [38,39,40].

**Table 2 pharmaceuticals-16-01299-t002:** Optical properties of symmetrical dyes from Figure 1. ^A^ DMSO ^B^ PBS. ^C^ MeOH.

Dye	λ_abs_ (nm)	λ_em_ (nm)	ε (M^−1^ cm^−1^)	Stokes Shift (nm)	φ_f_ (%)	Solvent
**1** [4]	624	633	168,600	9	-	PBS
**2** [4]	626	637	137,250	11	6	PBS
**3** [4]	633	638	157,300	5	9	PBS
**4** [4]	633	638	128,500	5	11	PBS
**5** [38]	636	653	149,000	17	32	PBS
**6** [38]	632	642	265,000	10	7	PBS
**7** [41,42]	656 ^A^	642 ^B^	295,000 [43] ^C^	8 [43] ^C^	12 [43] ^C^	
**8** [41,42]	659 ^A^	644 ^B^	-	-	0.8 ^B^	
**9** [39]	660	685	-	25	-	DMSO
**10** [38]	670	693	123,000	23	19	PBS
**11** [38]	667	685	188,000	18	7	PBS
**12** [38]	638	666	88,000	28	20	PBS
**13** [38]	627	650	150,000	23	5	PBS
**14** [41,42]	684 ^A^	669 ^B^	-	-	2.4 ^B^	
**15** [41,42]	686 ^A^	676 ^B^	-	-	0.1 ^B^	
**16** [44]	665	680	165,000	15	29.2	PBS

The optical properties of the symmetrical squaraine dyes have shown a wide range of varying characteristics, due to the change in the indole heterocycle or the modification of the squaraine core, as shown in Table 2 and Figure 1. When looking at the trends of the halogen effect in squaraines **1**–**4** [4], there are minimal to no changes seen in the absorbance and emission wavelengths. The difference in the molar extinction coefficients is ~40,000 M^−1^ cm^−1^, but there is no clear trend. Squaraines **2** and **3** break the down period trend. This indicates that halogens have minimal effects on the optical properties of the dyes. With regard to the addition of another benzene ring to the indole at the fourth and fifth positions, as in squaraines **14**–**15** [42], there is a noticeable red shift of around 25 nm in absorbance and emission compared to squaraines **7**–**8** [41]. A similar red shift is observed for squaraine **16.** The red shift in absorbance stems from the increased conjugation system of the dyes.

Another aspect of the indole heterocycle is that it can have modifications through *N*-alkylation. This modification brings changes to the molar extinction coefficient and the quantum yield of the dye. There are no considerable changes between absorbance and emission wavelength in norsquaraine dyes, with no alkyl substitution of the nitrogen atom of indole [41], as in squaraine dyes **5**–**6** and **10**–**13** [41]. There is a trend when considering the quantum yield of the same dyes: the molar extinction coefficient is higher for the dyes with *N*-alkylation, while the quantum yield is higher for the norsquaraine dyes when in phosphate-buffered saline (PBS). Additionally, the *N*-alkylation can be modified to include large groups such as the peptide, RGD (squaraine **16**). These modifications aid in improving the other properties needed for specific applications.

Another modification to the dye structure is changing the squaraine core. These modifications present different optical characteristics. The modification changes the central functional group on the squaraine core between a ketone, dicyanomethylene (squaraines **9**–**11)**, and barbituric acid (squaraines **12**–**13**). When observing the properties of dicyanomethylene squaraine dyes, there is a red shift observed, as seen in squaraines **5**, **6**, **10**, and **11** [38]. The difference seen is around 35 nm. The red shift is observed due to the dicyanomethylene group fixing the dye into the cis conformation [45]. This reduces that bandgap, resulting in the red shift [46]. When looking at the effect of barbituric acid on the optical properties, there is little change seen in the absorbance maxima between the ketone and barbiturate squaraine dye. However, there is some red shift in emission maxima allowing for a larger Stokes shift, as seen in squaraines **5**, **6**, **12**, and **13** [38]. Another change for the barbituric-acid-containing squaraine dye is in the molar extinction coefficient and quantum yield. The trend is a decrease in molar extinction coefficient and quantum yield when the barbituric acid modification is introduced.

The other class of squaraine dyes is unsymmetrical squaraine dyes. These dyes have unique properties compared to symmetrical dyes. Table 3 demonstrates the optical properties of these dye and Figure 2 shows the structure of these dyes.

Figure 2, as described previously, has similar changes observed to the dyes: modification to the *N*-alkylation of the indole heterocycle, changes of substituents at the fifth position of the indole, and modification to the squaraine core. The unique aspect seen for unsymmetrical indole-based squaraine dyes is the introduction of different indole or another heterocycle into the dyes. Adding another heterocycle or indole to the semisquaraine can introduce different properties, such as a new charged group, as in squaraines **19**–**24**, or change the conjugated system, as in squaraines **25**–**27**. These modifications allow fine-tuning of the properties of the dye.

Table 3 highlights the varying optical properties of unsymmetrical squaraine dyes. The data from Table 3 can be used to identify some trends. First, as mentioned earlier, the unsymmetrical squaraine dyes containing halogens, squaraines **19** and **21**–**23**, have no effect on the absorbance, emission maxima, quantum yield, and molar extinction coefficient. Takeshi Fukuda et al. reported the optical properties of the same dyes using PBS with 5% BSA. However, the quantum yield reported is significantly higher, ranging between 57% to 80%, and the emission maxima is reported to be around 650 nm [47]. An example is squaraine **20**, which has the most prominent quantum yield in Table 3. The dye contains water solubility modifications, which reduces the aggregate effect, and the addition of bovine serum albumin (BSA) can be attributed the low quantum yield. Considering the effect of different *N*-alkylated groups on unsymmetrical dyes, there is not much impact on the absorbance and emission maxima in squaraines **17**, **18**, **19**, and **24**.

Unsymmetrical squaraine dyes can have two different indole or other heterocycles. When examining the effect of adding an anilino group on at squaraine **25**, a large Stokes shift and a high quantum yield are observed [48]. When comparing squaraines **26** and **27**, with a larger heterocycle, the results show a larger increase in absorbance and emission maxima. Due to the varied abilities to donate electrons, the various donor groups in unsymmetrical dyes alter the optical properties. There is a decrease in the molar extinction coefficient of the unsymmetrical dyes compared to symmetrical dyes, with a slight blue shift in the maxima of absorbance and emission for the unsymmetrical dye. The quantum yield is comparable between the two scaffolds.

### 3.2. Applications of Indole-Based Dyes

Previous reports in literature have demonstrated the weak stability of the squaraine core. This is due to the reactivity of the squaraine core with the solvents and biological molecules, which results in the decomposition of the probe [50,51]. The biological nucleophiles, such as proteins, attack the squaraine core in vivo, which decreases the dyes biological usability. The squaraine core is attacked in vivo by biological nucleophiles, proteins that reduce the biological usability of dye. Attempts at improving stability have been made by adding external polymers [52], polymerization [53], and large macrocycles [50,51,54] to shield the core. The modifications reduce the yield and makes the synthetical process long and complicated. A new design of squaraines with quaternary ammonium cations on the alkyl chain was recently created by the Henary Group, the newly created structure has demonstrated the ability to stabilize the squaraine core. The stabilization is achieved by increasing the rigidity of the dye due to the electrostatic interactions between the negatively charged squaraine core and the ammonium cation. The stability of the probe is greater than a commercially available standard compound (Cy5) [4].

Based on the work from Henary’s group, it has been demonstrated that squaraine dyes with quaternary ammonium cations on the alkyl chain can be used for NIR fluorescence imaging [4]. Squaraine **21** (Figure 2) has shown the ability to image ovarian cancer. This dye can enter the tumor through an organic cation transporter mechanism (Figure 3E). The dye localizes in the lysosomes of the cell and has a high target-to-noise ratio and a long imaging window between 2 and 24 h post-injection [47]. The maximum signal-to-noise ratio is seen around 2.5 h after injection, as shown in Figure 3D. Figure 3A,B demonstrate the probe’s ability to identify more tumors that cannot be distinguished using color images.

Squaraine **25** (Figure 2) has shown the ability to image lipid droplets and endoplasmic reticulum in cells because of its lipophilic properties [48]. Lipid droplets have been reported to be linked with metabolic diseases and cancers [48,55]. Squaraine **25** initially stains the lipid droplet, and then the endoplasmic reticulum after 30 min, and the uptake into the cell occurs through diffusion [55]. The dye has a strong fluorescence signal in HeLa cancer cells.

Photoacoustic imaging is another method of bioimaging. Squaraine **16** (Figure 1) has shown a capacity for dual-imaging modalities: fluorescence imaging and photoacoustic imaging of tumors, as illustrated in Figure 4 [44]. The dye contains two groups of cyclic arginylglycylaspartic acid (cRGD), a known targeting agent for cancer, specifically angiogenic cells [44,52]. The cRGD group was added through a Cu(I)-catalyzed click reaction. The probe targets the α_v_β_3_ integrin, which is overexpressed in cancer [56] and gives the strongest fluorescence intensity 6 h after injection. The strongest photoacoustic imaging signal was observed between 680 nm and 700 nm, and was stabilized 2 h after injection [46].

Indole-based squaraine dyes can also be used as sensors. Sensors detect the presence of the analyte in a system. The detection of a single analyte can be achieved through multiple designs. Xiaoquin Liu et al. and Yuanyuan He et al. synthesized two squaraine dyes that can detect trace amounts of iron (III) ions in a solution through colorimetric and spectroscopic methods, as represented in Figure 5A–E [19,57]. Xiaoquin Liu et al. incorporate thymine moiety as an ion acceptor and bind to Fe^3+^ in 1:1 stoichiometry. Iron (III) concentrations as low as 1 µM in a 20% acetic acid-water solution were detected using Squaraine **28**. The dye, as well as utilized changes in absorbance, was used to determine the iron ion concentration; as the concentration increases, the absorbance intensity decreases at 635 nm [57]. The mode of action is a turn-off mechanism to determine the Fe (III) concentration in the solution, as shown in Figure 5B. Yuanyuan He et al. synthesized an unsymmetrical dye incorporating a fluorinated non-alkylated indole and a benzo[*e*]indole heterocycle. Squaraine **29** can be used to detect iron (III) and copper (II) ions through quantitative and colorimetric means (Figure 5E). The dye binds to both ions in a 2:1, dye: ion stoichiometry. The probe can be used to detect Fe (III) to approximately 3 µM and Cu (II) to 2.7 µM with a usable pH range between 2 and 8. The addition of iron (III) increases the fluorescence intensity but also results in a blue shift as concentration increases (Figure 5D) [19]. The mode of determining the concentration of copper is a turn-off absorbance mechanism; for iron, it is through a turn-on fluorescence mechanism (Figure 5C,D).

Guimei Wang et al. reported that squaraine **30** (Figure 6) can measure the concentration and image adenosine 5′-triphosphate, ATP, in cells [58]. The probe is a symmetrical, indole-based squaraine with a dicyanomethylene-modified squaraine core and is *N*-alkylated with phenylboronic acid. The dye forms a supramolecule in the solution that can be used to detect ATP. The boronic acid forms electrostatic interactions with ATP, allowing for qualitative and quantitative analysis. The mechanism is turn-on fluorescence; as ATP concentration rises, the fluorescence intensity also rises (Figure 6). The specificity of the probe is lacking, as there is a turn-on response for similar nucleotide structures: UTP, GTP, and CTP. The detection limit is 28 nM following 1:2, dye:ATP stoichiometry. Squaraine **30** can also be used to monitor ATP in a cell via fluorescence imaging [58]. The fluorophore can be used to investigate the functions and reactions of ATP and other energy molecules in response to outside influences like diseases or drugs.

Indole-based squaraine dyes have shown the ability to be used as protein sensors. Squaraine **13** (Figure 1) can bind to human serum albumin (HSA) and bovine serum albumin (BSA). Serum albumin concentration changes can be a biomarker for various diseases and disorders [59]. The probe has shown the ability to bind to all sites on HSA with a dissociation constant of 1.14 µM. The dye has a binding constant for HSA about three times higher than BSA, indicating a stronger binding affinity for HAS [60]. Once the fluorophore binds to serum albumin, fluorescence turns on, allowing for the quantification of the concentration [42,60].

Squaraines **7**, **8**, **14**, and **15** (Figure 1) can be used to detect various classes of proteins: transferrin, fibrinogen, trypsin, pepsin, and protease. Most to all squaraine dyes form aggregates in aqueous media, causing fluorescence quenching [41,42]. When proteins are added to the aqueous solution, the fluorescence increases due to the disaggregation and formation of the monomeric dye [42]. The turn-on fluorescence enables the determinination of the association and dissociation constants [42,60]. Squaraines **7** and **14** increase in molecular brightness when all the tested proteins are added to the PBS solution. Squaraines **8** and **15** only show an increase in molecular brightness when transferrin or fibrinogen is added to the solution. The surface hydrophobicity of the protein plays a role in the turn-on fluorescence mechanism for shorter alkyl chains. For longer alkyl chains, surface hydrophobicity and electrostatic interactions play a role in achieving turn-on fluorescence [42].

Xiaoxue Jing et al. reported that squaraine **9** (Figure 1) can be used to determine diquat in an aqueous solution. Diquat is an herbicide that has harmful effects on humans and animals after being ingested from contaminated water or crops. The probe follows a turn-on fluorescence mechanism, which is selective for only diquat. It does not show a significant increase in fluorescence when other ions or pesticides are used (Figure 7A). Adding diquat to a solution, like a protein, allows the dye to disaggregate, allowing the fluorescence to return because the probe returns to the monomeric form (Figure 7B) [39,42]. The return of fluorescence enables a visualization, via fluorescence imaging, of whether diquat has been ingested [39].

Indole-based squaraine dyes can be used as therapeutic agents to treat various diseases. Squaraine **31** (Figure 8) has shown the ability to be used as an anticancer and antibacterial agent. This is achieved because the probe is an effective photosensitizer for photodynamic therapy. The probe can generate reactive oxygen species (ROS), a known tool for killing cancer cells [61,62]. An important point is to minimize cytotoxicity when added to healthy cells. The probe needs to generate ROS when exposed to light to be an effective photosensitizer. Squaraine **31** has shown the ability to kill cancer cells, as presented in Figure 8A. Over 80% of the cells are killed in the presence of light when 25 μM of dye is introduced. The increase in cell viability above 100% in dark conditions is attributed to tumor cell growth. This indicates that the dye does not show cytotoxic activity in dark conditions. In addition to being an antitumor agent, the probe is an effective antibacterial agent. The dye utilizes photodynamic therapy to generate ROS species to eliminate Methicillin-resistant Staphylococcus aureus (MRSA), a drug-resistant bacteria. Figure 8B demonstrates how the probe operates in light settings and demonstrates dose-based effectiveness [62].

Eurico Lima et al. have synthesized six squaraine dyes that effectively generate ROS faster than the commercially available FDA-approved standard methylene blue [61]. Indole-based squaraine dyes can produce ROS. However, to be an effective photosensitizer, the dye should not be cytotoxic when added to healthy cells and should effectively kill cancer cells.

## 4. Quinoline-Based Squaraine Dyes

Quinoline is another heterocycle that can be incorporated into squaraines. Quinolines are aromatic cyclic molecules composed of benzene and pyridine moieties [63]. The two forms of the heterocycles that can be integrated into the dye are 2-methylquinoline (quinaldine) and 4-methylquinoline (lepidine) [64]. These quinoline compounds can be synthesized using aniline, as illustrated in Figure 4. Quinoline-based squaraine dyes are formed following the proposed mechanism shown in Figure 1. Quinoline, like indoles, can undergo *N*-alkylation to form various salts, quinoliniums, that can be used to form squaraine dyes.

Quinolines have been incorporated into a wide range of NIR dyes. Quinolines are a common heterocycle for other NIR scaffolds such as cyanine dyes. Quinoline-based cyanine dyes have numerous synthesized dyes with lots of applications [64]. In contrast, there are fewer reported structures and related applications for quinoline-based squaraine dyes.

### 4.1. Optical Properties of Quinoline-Based Dyes

The optical properties of quinoline-based squaraine dyes show a wide range of features. These are dependent on the modification within the dye. Below are the reported quinoline-based dyes in Figure 9, and their optical properties are shown in Table 4. The optical properties that will be evaluated are the absorbance maxima (λ_abs_), emission maxima (λ_em_), molar extinction coefficient (ε), quantum yield (φ_f_), and Stokes shift.

In addition to N-alkylation, modifications at the sixth or seventh positions of the heterocycle are common. Consequently, the modification can range from the addition of halogens (squaraines **32**–**35**), biological molecules (squaraines **36**–**37** and **44**), and other heterocyclic rings, as presented in Figure 9. These modifications introduce different structural elements into the dyes that can influence the optical properties or potential applications. In addition, quinaldine is a popular option compared to lepidine, used to form squaraine dye. Unsymmetrical dyes can be formed using two different quinoline heterocycles, squaraines **44** and **45**, or different heterocycles altogether, squaraines **40**–**43** and **46**–**47**.

In contrast to indole-based dyes, the majority of the absorbance maxima for quinoline-based squaraines are red-shifted. Squaraine **35** has the furthest red shift, it exhibits emission maxima at 992 nm and an absorption maximum at 900 nm. The probes with absorbance maxima at or near 900 nm are composed of lepidine and have a dicyanomethylene-modified squaraine core (squaraines **32**–**35**) [67,68]. This is to be expected, as the lepidine increases the conjugated system compared to quinaldine, resulting in a red shift absorbance [67,68]. The dicyanomethylene modification is known to red shift due to its electron-withdrawing characteristics and fixates the probe into the cis conformation [45,73]. When considering the quinaldine version of the dyes without a modified squaraine core, the absorbance maxima are seen to be around the mid-700s region (squaraines **36**–**39**). The quantum yield of these dyes is very low, even though organic solvents are used. This indicates that the absorbed energy is being relaxed by other pathways and fluorescence intensity is weak for these fluorophores based on the low quantum yield.

When different halogens are considered on the quinoline heterocycle, the modification causes the absorbance and emission spectra to red shift. Additionally, as shown in Table 4, the halogen modification causes the emission maxima to be red-shifted more significantly than the absorbance, with the largest effect being observed for the iodine-modified squaraine **35**. This effect results in a larger Stokes shift for the probe with iodine. The quantum yield for squaraine **35** is slightly higher than the other halogen modifications, while the lowest is for the chlorine modification, in squaraine **33** [68]. The molar extinction of squaraines **32** and **35** is the largest for the quinoline-based dyes; these probes containin iodine and hydrogen modifications [67,68]. Squaraine **34**, a probe modified with bromine, has the lowest molar extinction coefficient. The difference between the largest and the smallest molar extinction coefficient is 23,000 M^−1^ cm^−1^, which is not an extreme difference. The trend observed is that as atomic size of the halogen increases the absorbance and emission maxima, but the molar extinction coefficient and quantum yield remain about the same [68].

Squaraines **36** and **37** (Figure 9) incorporate biomolecules into the heterocycle with a significant difference in the molar extinction coefficient. When cholesterol is incorporated into the dye, the molar extinction coefficient increases compared to the glucose modification. A minor difference in the absorbance maxima, 6 nm, between the probes indicates that the modification has only a minor effect on the conjugated system. Jyothish Kuthanapillil et al. reported that fluorescence is above 765 nm for both structures due to the components contributing to the absorbance and the fluorescence being structurally similar [69]. Squaraine **44** differs from squaraine **37** in that quinoline is modified with an iodine group at the sixth position, creating an unsymmetrical structure. The modification resulted in the molar extinction coefficient being 70,000 M^−1^ cm^−1^ larger and the absorbance maxima, red shifted by 3 nm for squaraine **44** [69]. When altering the alkyl substituents on the nitrogen atom, as in squaraines **38** and **39**, the trend seen is the molar extinction coefficient being greater for the shorter alkyl chains. However, the absorbance for the longer chain, in squaraine **39**, exhibits a slight red shift [70].

Squaraines **40**–**43** are unsymmetrical dyes that use a benzothiazole heterocycle attached to the quinoline-based semi-squaraine dye with various modified squaraine cores. The modifications seen in the squaraine core involve various electron-donating groups. In comparison, squaraine **40**, which has not undergone any modifications, is blue shifted compared to some other modifications, such as those of squaraines **42**–**43**. The methylation of the oxygen in squaraine **41** resulted in the largest blue shift of 49 nm compared to an unmodified squaraine core dye. Additionally, the modification of the electron-donating group decreased the molar extinction coefficient. The probe with the amine modification, squaraine **42**, has the smallest molar extinction coefficient, with the difference being approximately 34,000 M^−1^ cm^−1^. The modification of the methylamine on the squaraine core in squaraine **43**, results in an increase in the singlet oxygen quantum yield compared to squaraine **40** [9,74].

Unsymmetrical dyes with distinctive structural configurations have been reported by Zhenxing Cong et al. [74] Squaraines **46** and **47** have heterocycles attached on the first and second positions of the cyclobutene ring. Usually, squaraine dyes have heterocycles attached to the first and third positions on the squaraine core. Squaraines **46** and **47** have the lowest absorbances and emission maxima from the set of dyes, which do not reach the NIR range. The absorbance and emission are seen in the visible region. In addition, the quantum yield is meager even though the optical properties were conducted in dichloromethane [72].

Many of the data on quinoline-based squaraine dyes’ optical properties are not published in the literature, indicating that the optical properties of this class of compounds need to be adequately assessed in future reported research. The reported optical properties are more thorough and easily accessible for indole-based squaraine dyes in the more recent works. Due to the limited data on quinoline-based squaraine dyes, there is a wide range of modifications available to develop a clearer picture of this class of dyes.

### 4.2. Applications of Quinoline-Based Dyes

The literature has explored the solar cell capabilities of these probes [13,71,75,76,77]. Quinoline-based squaraine dyes are optimal for dye-sensitized solar cell, as they have strong molar extinction coefficients [13,71,75,78], intense absorbance in the NIR range, and are stable [71,76]. These dyes transfer electrons from the high occupied molecular orbital (HOMO) to the lowest unoccupied molecular orbital (LUMO), and to the conduction band. When the dye is in an excited state, the electrons are injected into the conductor band [71,76,77]. Squaraine **48** (Figure 10) shows the ability to act as sensitizer for dye-sensitized solar cells (DSSCs). The dye has an absorbance maximum of 742 nm in ethanol and has a broad signal that is able to cover a wide range of wavelengths from 650 nm to 800 nm on a TiO_2_ thin film. The thermodynamics of electron injection favor HOMO and LUMO energy. The dye formes aggregates but has an open circuit voltage of 0.26 V and a low conversion efficiency of 0.53%. Squaraine **48** has a large IPCE maxima of 31%, as depicted in Figure 10 [76].

Squaraine **45** (Figure 9) has shown the ability to be a sensitizer for DSSC. The probe has a unique structure composed of a carboxylic quinaldine group linked to an indenoquinaldine by the squaraine core. The indenoquinaldine group extends the conjugation. The dye is thermodynamically favorable for electron injection. When squaraine **45** and 3α,7α-dihydroxy-5β-cholanic acid (CDCA) are added, the open circuit voltage is 0.576 V, and the conversion efficiency is 4.15%. The IPCE maxima value is around 55% near 800 nm (Figure 11A). The short circuit current density is nearly doubled when CDCA is added [71]. The use of the CDCA promotes the disaggregation of the dye, improving the photovoltaic properties, as shown in Figure 11B [71,78,79].

Quinoline-based squaraine dyes have shown the ability to be used as therapeutic agents. Squaraines **40**–**43** (Figure 9) have shown the ability to act as photosensitizers. These dyes have shown moderate photostability. However, squaraine **41** has the fastest photodegradation of the set. The dye loses about 80% of its absorbance after 60 min of light exposure. In contrast, the other dyes have shown a about 50% loss in absorbance after 60 min. All the dyes in the set show dose-dependent toxicity toward the Hep G2 and Caco-2 cancer cell lines. Squaraine **40** shows the lowest dark toxicity for both cell lines and low IC_50_ values in light conditions. Squaraines **42** and **43** show the lowest IC_50_ values when in light conditions in both cancer cell lines. The probes show toxicity in dark conditions, especially for the Hep G2 cell line [9].

Palasseri Sujai et al. [80] have reported a three-prong theragnostic agent that utilizes a symmetrical quinoline-based squaraine dye to act as a photosensitizer and detect uptake via multiplex Raman scattering, as depicted in Figure 12. The team has synthesized a nano envelope targeting metastatic melanoma through photothermal therapy, photodynamic therapy, and chemotherapy [80]. Squaraine **49** (Figure 12) incorporates a quinoline heterocycle with a sulfobutyl *N*-alkylation to improve water solubility, and iodine to enhance ROS production through the heavy atom effect [9,74,80,81]. The absorbance maximum is 754 nm, and the emission is 777 nm in DMSO. Due to the surface anti-DR5 antibodies, the dye was taken up by the tumor when it was incorporated into the nano envelope. The dye shows ROS generation when incorporated into the nano envelope and shows a 2-fold increase in ROS generation after being irradiated in the cell. The increased amount of DNA fragments after exposure to light causes an increase in cancer cell deaths [80].

Squaraines **46** and **47** (Figure 9) have shown the ability to be used for bioimaging. These dyes have fluorescence properties and have shown strong quantum yields in crystal form; squaraine **47** has a quantum yield of 42.45% in crystal form. The dyes show emissions in the NIR range when in crystallized form: 688 nm for squaraine **46** and 676 nm for squaraine **47**. Zhenxing Cong et al. have reported that these dyes are biocompatible and can be used to image cancer cells. The downside of these dyes is that the emissions are not in the NIR range in solution form [72]. This inhibits the dyes’ performance to obtain the region’s benefits: deeper tissue penetration and a strong signal-to-background for imaging [68,81,82].

Squaraine **50** (Figure 13) can be used to image using fluorescence and optoacoustic modalities to track and visualize lymphatic metastasis. The dye is unsymmetrical includes an indole and quinaldine heterocycle modified with a triarylamine containing a nitro group. In addition, the dye has a dicyanomethylene squaraine core. Squaraine **50** forms nanoprobe-aggregates in aqueous solution, enabling fluorescence and optoacoustic imaging [83]. As the probe contains a nitro group, it can react with nitroreductase (NTR), which is overexpressed in solid tumor hypoxia [83,84,85]. When the reaction between the probe and nitroreductase occurs, the nitro group is reduced to an amine group, as depicted in Figure 13. When this conversion occurs, the absorbance undergoes a blueshift from 740 to 690 nm (Figure 14A). The dye’s fluorescence and optoacoustic signal reemerge simultaneously, as shown in Figure 14B–D [85]. For optoacoustic imaging and fluorescence, the dye uses a turn-on mechanism.

The probe can image using optoacoustic mode through multispectral optoacoustic tomography (MSOT). This imaging method can be used to obtain 3D images by stacking the cross-sectional images from 2D optoacoustic images. Squaraine **50** was used to produced fluorescence imaging through aggregation-enhanced emission (AEE). This occurs when aggerated dye emits more intensely than in the monomeric form. The fluorescence and optoacoustic image were able to be visualized after 4 h and were present 48 h after injection (Figure 14E,F). Fluorescence imaging was used to determine that dye was excreted through the kidney, as presented in Figure 14F [83].

Bo Wu et al. [84] have reported a quinoline-based squaraine dye as a sensor to detect and image an enzyme: leucine aminopeptidase. Overexpression of leucine aminopeptidase (LAP) can indicate human diseases such as hepatitis [82], cholestasis [86], cirrhosis [87], and liver cancer [82]. Squaraine **51** (Figure 15) is an unsymmetrical dye that is composed of an indole that has a triethylene glycol chain to improve the water solubility and a quinoline that has a leucine amino acid group; these heterocycles are attached using an unmodified squaraine core. The leucine group on the quinoline heterocycle reacts with LAP, causing a cleavage at the amino group, as outlined in Figure 15A. When this cleavage occurs, the fluorescence of the probe increases at 710 nm and reaches maximum fluorescence after 30 min (Figure 15B,C). This probe is only selective for LAP and has a limit of detection of 0.61 ng/mL [82]. As there is an increase in fluorescence intensity, the mechanism of action the probe follows is a turn-on.

Squaraine **51** can be used to image LAP in vitro and in vivo. The probe can image the cancer cells, as LAP is overexpressed in cancers. When the dye is injected into tumor-bearing mice, the fluorescence gradually increases and becomes maximal after 30 min, as illustrated in Figure 15D. The probe has high intensity at the tumor site 4 h after injection. Due to the high concentration of LAP, the signal intensity is strong in liver tissue [82].

## 5. Perimidine-Based Squaraine Dye

Perimidine is a unique heterocycle due to its structure and electronic properties. It is a tricyclic system that contains two nitrogen atoms and has three six-member rings fused together [88,89]. It is comprised of a pyrimidine ring fused to a naphthalene group [88]. Due to its unique composition and connectivity, the heterocycle has a π-excessive and -deficient electron distribution [89]. This stems from the lone pair of electrons on the nitrogen atoms shifting the electron density towards the naphthalene group [89,90]. This allows the formation of a conjugated system and the unique properties of this class of dyes.

The synthesis of the perimidine heterocycle is achieved through many routes. The synthesis of perimidine can be performed quantitatively using green synthesis methods. The general reaction is achieved through a condensation reaction mixture with 1,8 diaminonaphthalene and a carbonyl derivate [89]. The two nitrogen groups on the heterocycle can be alkylated to further alter the heterocycle. (Figure 5). The dye is formed following the proposed mechanism shown in Figure 1 [37,91].

The perimidine heterocycle was first reported by de Aguiar in 1874, and since then, there has been lots of work regarding this heterocycle. There have been many applications regarding the uses of different perimidine variants in the biological field. The initial synthesis of perimidine-based squaraine dye was carried out by Griffiths et al. in 1993 [92]. Since then, a limited amount of work has been published regarding the application of these dyes. A greater amount work has gone into studying the structural properties of these dyes. This may be due to the patents on many dye derivatives, lack of knowledge of this class of squaraines, or a mistake by Griffiths et al. regarding the linkage of the heterocycle to the squaraine core [93].

### 5.1. Optical Properties of Perimidine-Based Dyes

This class of squaraine dyes shows a unique set of properties are not seen by other heterocycles. Below are selected dyes in the literature (Figure 16) and their optical properties (Table 5). The optical properties that will be evaluated are the absorbance maxima (λ_abs_), emission maxima (λ_em_), molar extinction coefficient (ε), quantum yield (φ_f_), and Stokes shift.

A common modification of the perimidine heterocycle is having a long alkyl chain at the second position of the heterocycle. This is in contrast to the other classes of squaraines where the alkyl chain is seen from *N*-alkylation in heterocycles. Additionally, as there are four nitrogen atoms in the symmetrical dye as a result, there are four locations where *N*-alkylation can occur, as seen in squaraines **55**–**57**. This class of squaraines can form symmetrical and unsymmetrical versions, as seen in Figure 16. For the unsymmetrical versions, an indole heterocycle is incorporated into the dye, as in squaraines **58**–**61**.

Many dyes exhibit incomplete optical characterization of the vital optical properties shown in Table 5. This is due to the limited selection of literature regarding this class of squaraines. This is surprising even though there is a large redshift in absorbance compared to indole and quinaldine-based squaraine dyes. In addition, the reported probes have not explored the modified squaraine core that has been shown to red shift the absorbance and change the optical properties of the other classes of squaraines. When reflecting on the data of these dyes, there is a noticeable change in optical properties when the probes have *N*-alkylation. Squaraines **55** and **56** significantly increase in quantum yield compared to squaraine **52**. The only difference between squaraines **52** and **55** is the methylation of the nitrogen groups [37,94]. The addition of the alkyl chain on the nitrogen atom increases the fluorescence properties of these probes, leading to a higher quantum yield. The non-alkylated dyes can absorb light. However, they release that energy via non-radiative pathways; the main pathway of relaxation is through hydrogen-bond-induced intersystem crossing [94]. Looking at the molar extinction coefficient between squaraines **52**–**56**, there is an increase in the molar extinction coefficient when the *N*-alkylation is present [37,92,94]. The addition of alkyl chains on the nitrogen makes the fluorescence properties like that of indole- or quinoline-based squaraine dyes [9,44,47]. A downside of *N*-alkylation is a blue shift in the absorbance and fluorescence. There is a 65 nm blue shift in the absorbance between squaraines **52** and **55**. However, the blueshift in absorbance is not as significant between squaraines **54** and **57** (14 nm). This indicates that other factors play a role in the degree of blueshift that is observed.

Squaraines **58**–**61** are unsymmetrical dyes with a perimidine, while containing an indole, and benzo[*e*]indole heterocyle linked with a squaraine core [95]. Compared to symmetrical dyes, the unsymmetrical dyes follow a trend wherein there is a blue shift observed for absorbance and fluorescence. The unsymmetrical dyes have larger Stokes shifts than the symmetrical version (squaraines **52**, **55**–**56**). Additionally, squaraines **58**–**61** have a lower molar extinction coefficient. These properties are ditinct from those observed with unsymmetrical indole- and quinoline-based squaraine dyes.

Contrary to expectations, the benzo[*e*]indole-based squaraine dyes exhibit a red shift in absorbance when compared to the indole-based squaraine dyes, which is due to an increase in conjugation from the benzo[*e*]indole heterocycle [41,42]. However, no significant change is seen in the absorbance maxima between squaraines **58** and **60**, **59** and **61**. Another surprise is that the quantum yield for these dyes is almost nonexistent; this indicates that the perimidine moiety influences the optical properties to a far greater extent than the indole and its derivative [95].

The full scope of the optical properties of perimidine-based squaraine dyes cannot be determined. However, some initial conclusion can be made from the small set of optical properties. The first conclusion is that the non-alkylated version has an absorbance maximum around 800 nm and a moderate molar extinction coefficient, but no definite conclusions regarding the fluorescence and quantum yields can be made [92,94]. Another conclusion is that when the perimidine moiety is alkylated, there is a blue shift in absorbance and fluorescence. However, there is a significant increase in quantum yield [37,94]. When perimidines are included in unsymmetrical dyes, the perimidine has a greater role in influencing the optical properties [95].

### 5.2. Applications of Perimidine-Based Dyes

To demonstrate the actual linkage of the perimidine-based squaraine dyes, Mistol et al. published a work in 2015 [93]. As mentioned above, the reported linkage when the dye was published initially shows the perimidine bound to the squaraine core at the sixth position [92]. However, in the early 2000s, there was a patent filed by Imation Corp. that reported the dye linkage to be different from the work of Griffiths et al. [92,96]. Again in 2008, a paper presented the linkage as originally reported by Griffiths et al. [92,97]. The linkage of the non-alkylated dyes was shown to be connected at the four positions; 2D NMR proved this. The signal generated from COSY and HBSC indicates that the linkage occurs at the four positions of the perimidine because of the observed hydrogen bonding signal. The dye can form at the sixth position of the perimidine, but this would be a minor product in the reaction [93].

Kim Sung-Hoon et al. demonstrated the stability of these dyes. The photostability of squaraines **53**, **62**, and **63** (Figure 16 and Figure 17) were compared, and the stability of the dye was determined to be **63** > **53** > **62** (Figure 17). The probes that underwent the greatest degradation still held 40% of their initial absorbance, namely, squaraine **62** after 200 min of exposure to light conditions. The hydroperimidine-containing dye, squaraine **63**, has shown a red shift in absorbance, 858 nm, compared to the dihydroperimidine-containing probes, squaraines **53** and **63**. Additionally, squaraine **63** has shown better stability compared to squaraines **53** and **63**, losing about 20% of its original absorbance intensity. These probes show better photostability compared to other classes of dyes [98].

Perimidine-based squaraine dyes have shown the ability to be used as sensors. Squaraine **64** (Figure 18) shows the ability to be a single-time ammonia sensor. This dye has an absorbance maximum of 806 nm. The dye’s sensitivity to ammonia is solvent-dependent. When the probe is in ethanol and ammonia is introduced into the solution, there is no change in the absorption spectra. In contrast, when acetone and ammonia solution is used, the absorption peak decreases, as depicted in Figure 18. Attempt to regain the absorption peak have been made through adding HCl, but the peak was not generated, signifying that the sensing is irreversible. The dye undergoes a reaction with ammonia that leads to the decomposition of the dye and a decrease in the intensity of the peak. Additionally, the absorption peak decreases when only HCl is added to the dye. The inability to regain absorption is a downside of this dye in senssing ammonia [99].

Squaraine **57** (Figure 16) has shown a capacity for sensing bovine serum albumin (BSA). The probe contains four sulfonate groups, allowing the dye to be fully dissolved in a PBS buffer. When BSA is added, the fluorescence of the dye increases about 25-fold. The increase in fluorescence is due to the decrease in local hydrophilicity and the reduction of aggregate-induced quenching. The sensing mechanism is a turn-on fluorescence mechanism and can give an emission maximum above 800 nm. However, more information regarding the sensitivity and specificity of this probe, as seen for the other heterocycle-based dyes [37], is still needed.

Squaraine **53** (Figure 16) issued as an optical sensor. When stimulated by a chemical signal, this dye has been integrated into a PVC membrane to emit optical signals. Squaraine **53** has good solubility and stability, is clear on the PVC membrane, and can be protonated and deprotonated without a loss of optical signal. The downside is that the extraction of analytes on the membrane is not reversible, due to low basicity [100].

Squaraines **58**–**61** (Figure 16) have shown the ability to partition into the liposome bilayer. Once the dye is partitioned into the liposome membrane, the fluorescence turns on. Squaraines **59** and **61** have shown the strongest and fastest fluorescence after being introduced into the liposome environment. Squaraine **59** had the best performance in producing the most intense fluorescence signal after 30 min. Squaraines **58** and **60** were the fastest to reach maximum fluorescence, but the intensity was lower than that of other probes [95].

Squaraine **65** (Figure 19) shows the ability to be a tackifier. A tackifier is a substance used to improve the adhesion properties of another substance. The dye contains an abietic acid moiety; this group is utilized because it is a parent compound of a known tackifier, rosin acid. The absorbance for the probe is 810 nm in THF (Figure 19A). When the dye is added to different polymer films, the absorbance band broadens due to aggregation, as shown in Figure 19B [97]. The dye was able to act as an NIR hot-melt adhesive.

Perimidine-based squaraine dyes have been the subject of numerous patent applications over the years. The first patent was filed in 1993 by the 3M (Minnesota, mining and manufacturing) company, Saint Paul, MN, USA. This initial patent uses this class of squaraines for thermal imaging transfers. This patent is intended for use of the dyes as colorant donor elements. The dyes can convert light energy to thermal energy, which can heat the whole matrix or a localized section of it. This would enable the transfer of the color/image to occur in the designated area. Multiple perimidine-based squaraine dyes are claimed in the patent to be thermal imaging transfer agents [101].

## 6. Conclusions

Squaraine dyes are a popular class of NIR dyes with a wide range of optical properties and applications. Much work has gone into synthesizing squaraine dyes that utilize indole moieties. About half of the dyes reported include an indole moiety in their structure. The indole-based squaraine dyes have a high quantum yield and molar extinction coefficient, but their absorbances are capped near 700 nm. The indole-based squaraine dyes with absorbances above 700 nm contain various modifications, including a modified squaraine core and a large conjugated heterocycle. However, attempts to utilize indole-based squaraine dyes have been made in order to be used for various applications. In addition, in vivo stability has been improved by adding a quaternary ammonium cation group.

In comparison to indole, the other heterocycles, quinoline and perimidine, have fewer reported dyes. This is the case even though they have shown absorbance to be red-shifted to 900 nm. In addition, the quinoline- and perimidine-based squaraine dyes show strong molar extinction coefficients over 100,000 M^−1^ cm^−1^, and few dyes have shown a high quantum yield. Due to the limited exploration of these heterocycles, there are a limited number of dyes to analyze to determine how structural components impact the optical properties of these dyes. Based on this information, there are gaps in the understanding of how different structural components may lead to similar or different optical properties compared to indole-based dyes. As a result, assumptions about the limited number of dyes, regarding their optical properties, must be made.

Quinoline-based squaraine dyes have shown suitability for various applications. These applications are like those of indole-based squaraine dyes, with the added benefit of being red-shifted. Quinoline-based squaraine dyes are a great alternative to the indole variant, with much room for development.

The number of reported applications for perimidine-based squaraine dyes is nearly zero compared to the other heterocycles. This heterocycle is still a blank slate, with very few reported applications for biological and medical fields. The perimidine-based dyes have shown promising opportunities for the introduction of various modifications, and this could maximize their effectiveness in various applications. Hopefully, this report will foster the development of quinoline- and perimidine-based squaraine dyes.

## Data Availability

Data sharing is not applicable.

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
