# Peer review of "Exploration of NIR Squaraine Contrast Agents Containing Various Heterocycles: Synthesis, Optical Properties and Applications"

_pharmaceuticals, 2023, doi:10.3390/ph16091299_

Round 1
Reviewer 1 Report
See attached file.

See Attached file.
Author Response
We want to thank the Editor (Reviewer 1) for their kind remarks regarding the content of the review article. We have taken into consideration the feedback received in the editorial commentary and have made the appropriate changes to improve the language and readability of the manuscript. see corrected manuscript attached.

Reviewer 2 Report
Dear Authors:
This paper, presented a very interesting report for Exploration of NIR Squaraine Contrast Agents Containing Various Heterocycles: Synthesis, Optical Properties and Applications. The overall structure and writing indeed require a significant improvement. I really would like to see this article in Pharmaceuticals after major revisions.
1. You just mentioned general synthesis of squaraine dye. Provide some examples of synthetic methods that refer to the synthesis of derivatives of these compounds in the introduction section.
2. Provide explanations about the reason for choosing this dye for a review in the introduction section.
3. Correct the text written in line 145.
4. Figure 5. In parts B and E, add the name of the solvent. Also, Figure 6 and ….
5. Provide abbreviations in footnotes. Such as μM.
6. It is better to add the synthesis mechanism of these derivatives to the article.
Best Regards
Author Response
See attached letter.

Round 2
Reviewer 2 Report
Accept in present form.